# A Previously Unrecognized Granulomatous Variant of Gamma-Delta T-Cell Lymphoma

**Tatsiana Pukhalskaya** [1,*], **Bruce R. Smoller** [1], **David M. Menke** [2] **and Olayemi Sokumbi** [3]

1 Department of Pathology and Laboratory Medicine, University of Rochester Medical Center, Rochester, NY 14642, USA; bruce_smoller@urmc.rochester.edu
2 Department of Laboratory Medicine and Pathology, Mayo Clinic, Jacksonville, FL 32224, USA; menke.david@mayo.edu
3 Department of Dermatology, Mayo Clinic, Jacksonville, FL 32224, USA; ysokumbi@gmail.com
* Correspondence: tatsiana_pukhalskaya@urmc.rochester.edu

**Abstract:** Primary cutaneous γδ T-cell lymphoma (PCGD-TCL) is an extremely rare and aggressive T-cell neoplasm with complex heterogeneity. We present a series of two patients who presented with firm, subcutaneous nodules and were diagnosed with PCGD-TCL. In both cases, biopsies demonstrated a both superficial and deep adnexotropic infiltrate comprised of angiocentric, medium- to large-sized atypical lymphocytes. The infiltrate extended into the panniculus. Immuno–histochemical stains highlighted atypical lymphocytes that expressed CD3, CD8 and CD56 but were negative for EBV ISH. A brisk histiocytic response with focal aggregation into granulomas was highlighted with a PG-M1 stain. The atypical lymphocytes were positive for gene rearrangements on a TCR delta stain and negative for βF-1. CT and PET scan in one of the two patients demonstrated diffuse, subcutaneous, ground-glass foci; hypermetabolic soft tissue nodules; and lymphadenopathy in the lungs, as well as splenomegaly. A diagnosis of histiocyte-rich PCGD-TCL was rendered. A histiocyte-rich, granulomatous variant of γδ T-cell lymphoma is extremely rare. Its potentially misleading resemblance to inflammatory granulomatous conditions could pose a diagnostic pitfall in this already challenging condition. This variant may resemble granulomatous mycosis fungoides and granulomatous slack skin syndrome, but it has a distinct, aggressive clinical outcome.

**Keywords:** primary cutaneous γδ T-cell lymphoma; PCGD-TCL; histiocyte-rich PCGD-TCL





## 1. Introduction

Primary cutaneous γδ T-cell lymphoma (PCGD-TCL) is an extremely rare and aggressive T-cell neoplasm with clinical and pathologic heterogeneity and diagnostic complexity [1]. It was previously regarded as a variant of subcutaneous, panniculitis-like T-cell lymphoma (SPTCL), but was separated from this group because of significant clinical and phenotypic differences [2]. PCGD-TCL represents less than 1% of all lymphomas [1,3–5].

Clinical presentations of PCGD-TCL can be highly variable, with common presentations including ulcerated plaques and subcutaneous nodules on the upper and lower extremities. There may be accompanying involvement of mucosal and other extracutaneous sites [6]. Similar to SPTCL, PCGD-TCL might be accompanied by hemophagocytic syndrome but with a comparatively more aggressive clinical course. However, a slower-paced form of the disease has also been described [7,8]. Histologically, some cases of PCGD-TCL predominantly involve the epidermis and/or the dermis, while others are centered in the subcutaneous adipose tissue [9]. PCGD-TCL is commonly resistant to all forms of treatment, although cases with an indolent clinical course have been reported [1,10,11].

Histologically, PCGD-TCL is characterized as a multi-leveled cutaneous infiltrate composed of medium-to-large activated γδ T-cells with a cytotoxic phenotype. Involvement of the epidermis is frequent as well as infiltration of the dermis and subcutis. These lymphoma cells are clonal in nature and most commonly express granzyme B, TIA-1

and perforin [12,13]. Angiotropism and necrosis can also be present [14]. In most cases, immunohistochemical studies show the malignant cells to typically have the following phenotype: CD3+, CD4−, CD56+, EBV and variably, CD8+ [15]. These cells are negative for β-F1, a marker for the αβ phenotype, and can be distinguished by positive staining for TCR-γ or TCR- δ [16]. Staining for TCR-δ and TCR-γ using immunoperoxidase techniques can be conveniently performed in formalin-fixed, paraffin-embedded tissue sections. [15]. The staining for TCR-δ gives more reliable results then the one for TCR-γ.

Histiocyte-rich presentations of PCGD-TCL are extremely rare and may represent a diagnostic pitfall, thereby delaying timeliness and potential effectiveness of therapy. Herein, in this study we report two cases of a histiocyte-rich and focally granulomatous variant of PCGD-TCL.

## 2. Case Presentation

### 2.1. Case 1

A 28-year-old male without any significant past medical history was seen with a complaint of persistent skin nodules on the trunk and extremities. His symptoms presented 4 years earlier with concurrent spiking fever, nausea and emesis. His initial symptoms resolved for 6 months and subsequently recurred in association with generalized weakness, arthralgia, chest pain and multiple firm, subcutaneous nodules. The nodules were non-tender, tended to ulcerate and were self-resolving. Original histopathology performed at an outside institution was suggestive of lupus profundus, leading to the initiation of azathioprine therapy despite persistently negative lupus serology. At that time, imaging studies failed to reveal an ongoing systemic process. Although the patient noted the improvement of his fever, the nodules persisted and progressed after one year of treatment, leading to the initiation of methotrexate and oral prednisone. His clinical examination was notable for erythematous and indurated subcutaneous nodules scattered throughout his arms and legs as well as an erythematous and indurated plaque with central necrotic crust on the back (Figure 1). Histopathology of two different nodules demonstrated similar findings, revealing an epidermotropic, dermal and subcutaneous infiltrate comprised of medium-to-large, atypical lymphocytes and numerous histiocytes forming granulomas (Figure 2A–C). There was notable angiodestruction, necrosis and syringotropism. Immunohistochemical stains showed the atypical lymphocytes to be positive for CD3, CD8, CD56 and TCR δ with loss of CD7 and diminished expression of CD5 (Figure 2D,F,H). These cells were negative for CD4, CD20, CD30, EBV in situ hybridization, CD34, CD117, MPO and TdT (Figure 2E). Beta F1 was negative in the malignant cells while highlighted in the benign CD4 T-lymphocytes (Figure 2I). The histiocytes appeared to be of non-Langerhans type and expressed CD63 (PGM1) and CD163 (Figure 2G). CD68/163 cells represented around 70% of the malignant infiltrate. The presence of granzyme and TIA confirmed the cytotoxic nature of the infiltrate. A T-cell gene rearrangement study was additionally performed and confirmed a clonal T-cell population. In light of the clinical presentation, histopathologic features and immunophenotype, the neoplasm was classified as histiocyte-rich cutaneous γδ T-cell lymphoma. PET–CT scan revealed numerous hypermetabolic subcutaneous nodules, bilateral pulmonary nodules, splenomegaly and heterogeneity of metabolic activity within the bone marrow. Histological evaluation of the bone marrow was negative. The cutaneous manifestations preceded the systemic involvement by years; hence, hepatosplenic γδ T-cell lymphoma was excluded. The patient was referred to oncology and underwent 1 cycle of CHOEP (cyclophosphamide, doxorubicin, vincristine, prednisone and etoposide) chemotherapy with minimal response to therapy. He unfortunately died from complications of his disease 6 months after diagnostic confirmation and 4.5 years after the original presentation.

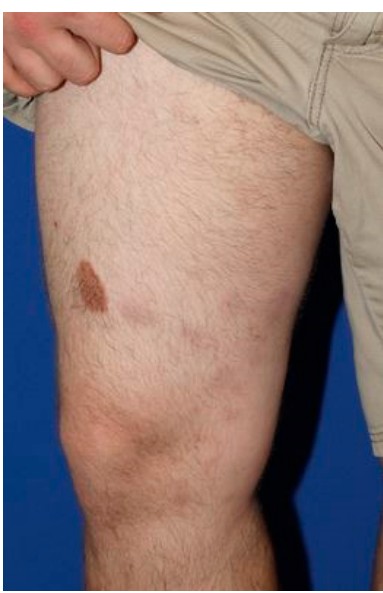

**Figure 1.** Clinical presentation of the patient from case#1, illustrating characteristic subcutaneous nodules.

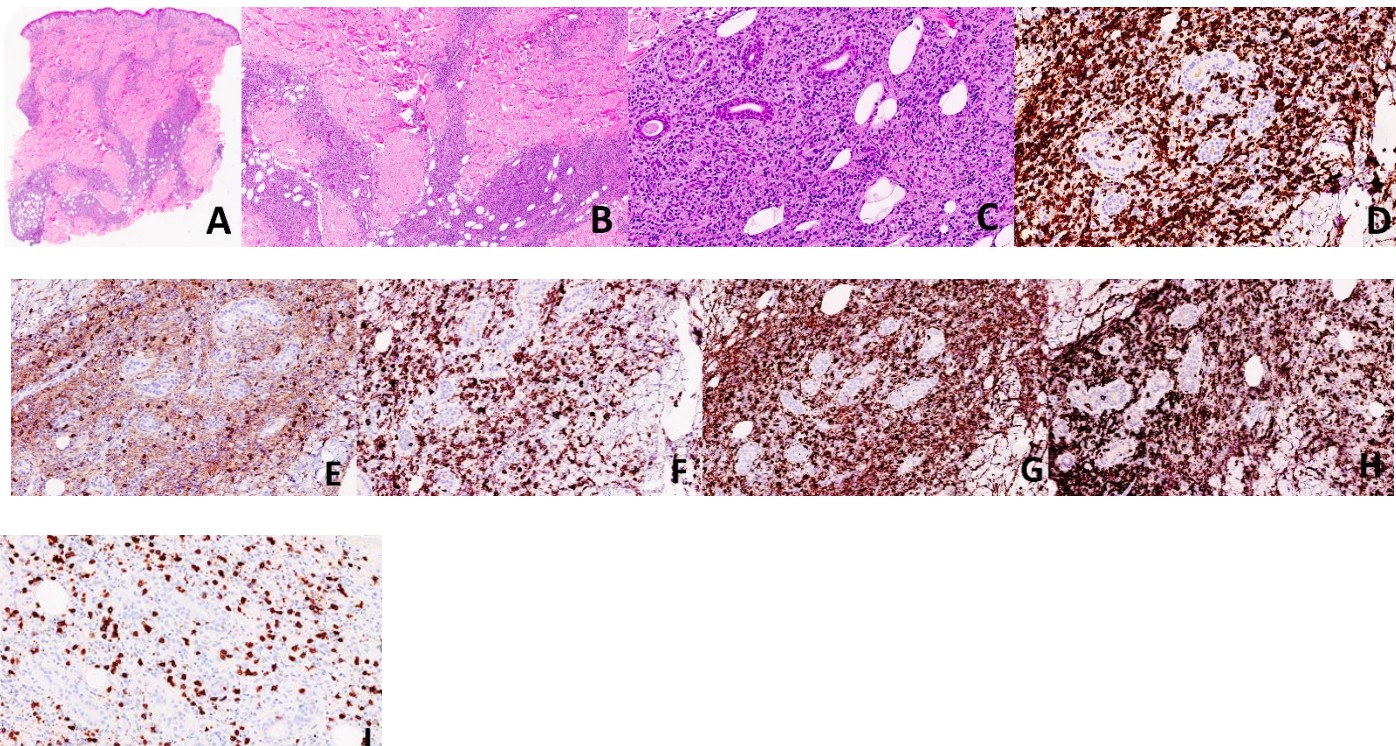

**Figure 2.** Case 1. Histology from the patient's lesion demonstrates an epidermotropic, dermal and subcutaneous infiltrate comprised of medium-to-large, atypical lymphocytes and numerous histiocytes forming granulomas (**A–C**: H&E, ×20, ×40, ×100, respectively). The atypical lymphocytes were highlighted by CD3 (**D**, ×200), CD8 (**F**, ×200), TCR delta (**H**, ×200), while they were negative for CD4 (**E**, ×200) and BF-1 (**I**, ×200). Histiocytes are highlighted by CD68 (**G**, ×200).

### 2.2. Case 2

A 66-year-old female presented with a 6-month history of a progressive lump behind her right knee. The lesion was clinically asymptomatic and was concerning for a cyst. Examination revealed a firm, erythematous, non-tender subcutaneous nodule. MRI image revealed a 2.6 by 2.4 cm, ill-defined enhancement within the subcutaneous fat with adjacent

skin thickening. Histopathological examination of the lesion revealed a predominantly subcutaneous infiltrate comprised of a variable mixture of small-to-medium sized, atypical-looking lymphocytes and abundant histiocytes forming small granulomas (Figure 3A–C). Occasional mitotic figures were noted. Immunohistochemical (IHC) markers revealed the atypical lymphocytes were exclusively T-cells that expressed CD3, CD8, CD56, granzyme B, TIA-1 and TCR γ while being negative for CD4, CD5, BF1 and CD30 (Figure 3D–F,H). CD68 highlighted abundant histiocytes (Figure 3G). CD68-positive cells represented 70–80% of the malignant infiltrate. EBV ISH showed strong nuclear staining in approximately 50% of the lymphocytic population. Flow cytometry demonstrated that the atypical lymphocytes lacked CD7, while DNA from the specimen revealed the presence of a clonal rearrangement of the T-cell receptor gamma gene. An infectious etiology for the granulomas was excluded, with special stains for organisms. There was no evidence of hemophagocytosis or angiodestruction. Overall, the morphology and immunophenotype of the lesion were consistent with primary cutaneous gamma-delta T-cell lymphoma, and the granulomas were thought to be an unusual histologic variant related to the process. PET scan showed no involvements of other organs. The patient was referred to oncology and underwent 3 cycles of ICE (ifosfamide/carboplatin/etoposide) chemotherapy, plus several treatments with radiation. Despite therapy, her disease progressed and developed multiple subcutaneous nodules. She died from complications of her disease after choosing to pursue palliative care.

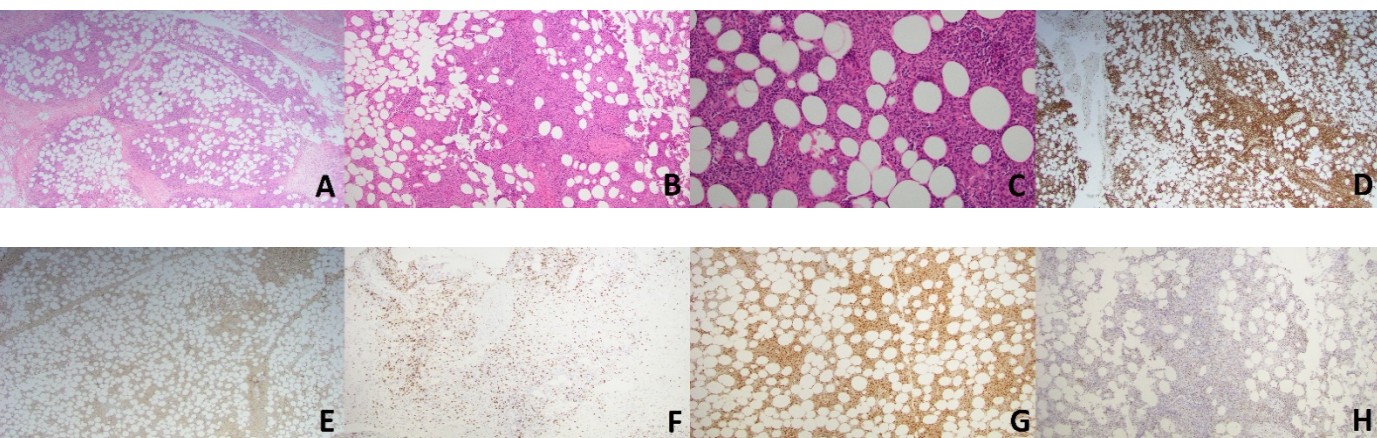

**Figure 3.** Case 2. Histology from the patient's lesion demonstrates predominantly subcutaneous infiltrate comprised of small to medium atypical lymphocytes and numerous histiocytes forming small granulomas (**A–C**; H&E; ×20, ×40, ×100, respectively). The atypical lymphocytes were highlighted by CD3 (**D**, ×20), CD8 (**F**, ×40) and TIA-1 (**H**, ×40) while they were negative forCD4 (**E**, ×20). Histiocytes are highlighted by CD68 (**G**, ×40).

## 3. Discussion

Our cases represent example of diagnostic challenges, not only due to the extreme rarity of the disease, but also because of their misleading and overall unusual histological pictures. Both cases featured primary cutaneous PCGD-TCL, in which microscopy showed atypical lymphocytes surrounded by significant amounts of histiocytes forming granulomas and thus obscuring the actual nature of the disease.

Although granulomas may be found in Hodgkin's and non-Hodgkin's lymphoma, they are extraordinarily rare in primary cutaneous lymphomas [17,18]. Exceptions include the "granulomatous" variant of mycosis fungoides (MF), granulomatous slack skin syndrome and rare cases of subcutaneous, panniculitis-like T-cell lymphoma (SPTCL) [19–22]. Regarding PCGD-TCL, Caudron et al. described granulomatous infiltrate in a case of PCGT-TCL that was initially misdiagnosed as an inflammatory panniculitis [23]. Scarabello et al. described cutaneous granulomas in a subcutaneous "panniculitis-like" variant of T-cell lymphoma, with cells derived from gamma-delta origin, CD4⁻, CD8⁺ and CD56⁺ [19].

The author reported a predominance of large clusters of giant cells of the Langhans and foreign body types, located mostly at the periphery of the fat lobules. In contrast to this earlier description, our cases demonstrated mostly sheets of polygonal histiocytes and small granulomas scattered throughout the lesion and not localized to the periphery of fat lobules. While an additional case report of PCGD-TCL described the presence of a histiocyte-rich infiltrate in the skin, it mainly focused on the florid granulomatous reaction in the bone marrow biopsy, suggesting the bone marrow finding to be an accompanying feature of the patient's primary cutaneous disease [24].

Mycobacterial and fungal infections as well as sarcoidosis and other cutaneous granulomatous processes should be excluded during the diagnostic pursuance of PCGD-TCL. Although rare, erythema nodosum leprosum (ENL) might similarly present with disseminated subcutaneous nodules and histology revealing a dermal infiltrate of neutrophils and lymphocytes superimposed on a collection of histiocytes [25]. In both our cases, special stains for organisms (AFB and GMS) were negative and in conjunction with the overall histopathology that supported the non-infectious nature of the process.

IHC analysis in our Case 2 showed that half the malignant cells stained with EBV. This is an uncommon observation in PCGD-TCL. Such a finding further complicated the diagnostic dilemma and brought the necessity of excluding NK/T-cell lymphoma nasal type (ENKTL) as well as secondary cutaneous involvement by other variants of $\gamma\delta$ T-cell lymphoma. Both ENKTL and non-cutaneous forms of $\gamma\delta$ T-cell lymphoma are known to express EBV [26]. ENKTL commonly originates in the upper respiratory tract, but involvement of other organs is frequent, particularly the skin, where the disease might be primary [27]. Cutaneous lesions of ENKTL present as erythematous or violaceous plaques and nodules. The histology might closely resemble subcutaneous panniculitis T-cell lymphoma or mycosis fungoides in addition to PCGD-TCL [27]. Nevertheless, neoplastic cells of ENKTL are characterized by an NK-cell phenotype and lack of T-cell markers, i.e., TCR-alpha, -beta, -gamma and -delta as well as markers for CD3, CD4, CD5 and CD8. CD56 and cytotoxic proteins (TIA-1, granzyme B and perforin) are positive in the majority of ENKTL cases, although CD56 is not unique to this form of lymphoma. In Case 2, the atypical lymphocytes were exclusively T-cells with strong expression of CD3, CD8, CD56, granzyme B, TIA-1 and TCR $\gamma/\delta$ as well as expression of CD7, a common marker of ENKTL. In addition, laboratory and PET data during the initial evaluation confirmed the cutaneous confinement of the disease, ruling out the possibility of its secondary involvement.

Furthermore, a literature search revealed a report of suspected PCGD-TCL with EBV expression [23,28]. Therefore, we believe that our finding adds to the body of literature. With an increasing number of reports, it might be more feasible in the future to separate the abovementioned phenotype as EBV T-cell lymphoproliferative disorder with gamma/delta phenotype. The possibility of an association between EBV and an increased number of histiocytes in PCGD-TCL is not clear and remains to be elucidated.

The rarity of PCGD-TCL can be correlated with the cells of origin, which represent between 0.5% and 16% of the body's T-cells [29]. There are three V$\delta$ segments, but the vast majority of $\gamma\delta$ T cells express either the V$\delta$1 or V$\delta$2 T-cell receptors, creating two distinguished subtypes of these cells: V$\delta$1 and V$\delta$2 [30]. Regarding types of $\gamma\delta$ T cells found in the skin, recent findings by Jay Daniels et al. reported PCGD-TCL to be derived from both V$\delta$1 and V$\delta$2 cells [31]. They proposed that $\gamma\delta$1 T-cells predominated in the epidermis and dermis and gave rise to lymphomas originating in the epidermis and dermis, while panniculitis-based lymphomas arose from V$\delta$2 cells. In Case 1, the tumor appeared as an epidermotropic, dermal and subcutaneous infiltrate, while in Case 2, it was predominantly panniculitic in distribution. The difference in the initial clinical presentations (multiple subcutaneous nodules/plaques, constitutional symptoms and slow course of the disease in Case 1 compared to single subcutaneous nodule without other symptoms and fulminant progression in Case 2) might support the possibility that these tumors originated from different $\delta$T-cell subtypes.

It is important to add that like αβ CD4+ T cells, γδ T cells can express abundant effector cytokines [29]. Similar to αβ CD8+ T cells, a γδ T cell can also express cytotoxic enzymes such as granzymes and perforin, which cause local and systemic inflammatory reactions [32,33]. These cells are able to recognize a broad range of antigens without the presence of major histocompatibility complex molecules as well as to attack target cells directly through their cytotoxic activity or indirectly through the activation of other immune cells [34]. γδ T cells are an important early source of the inflammatory cytokines interferon-γ (IFN-γ) and tumor necrosis factor (TNF)-α in many infections and other disease models [35]. Additionally, these cells are capable of serving as antigen-presenting cells by processing a wide range of antigens for presentation and by stimulating other immune cells [36,37]. γδ T –cells induce dendritic cell (DC) maturation through TNF-alpha. They induce the production of IL-12 by DC, an effect that involves IFN-γ production [38,39]. Petrasca et al. reported that δ2-T cells induced the expression of CD86 and HLA-DR and the release of IFN-γ, IL-6 and TNF-α by DC, and these stimulated DC proliferation and IFN-γ production by conventional T-cells [40]. Before γδ T-cell lymphoma was recognized as a separate entity, a subcutaneous, δ-positive T-cell lymphoma was described that produced high quantities of IFN-γ [41]. This reported patient presented with recurrent, spontaneously regressing, nontender subcutaneous nodules with a histology of SPTCL. Although Burg et al. did not identify large amounts of histiocytes or granulomas in the lesion, the clinical picture was very similar to our case and suggested that IFN-γ could be one of the main cytokines, if not the sole one, creating such symptoms. Therefore, we hypothesize that malignant γδ T –cells are activated by an unknown factor or acquire an ability to auto-activate and progress to stimulating a DC reaction, which subsequently attracts tissue histiocytes. These histiocytes are stimulated and produce IL-12, which triggers IFN-γ production by tumor cells. It is unclear in which subset of γδ T-cells this function might predominate. We believe that our report represents a practical illustration of the importance of maintaining a high level of suspicion when evaluating an unusual granulomatous dermatitis that may conceal the presence of an underlying PCGD-TCL. Nevertheless, the possibility of γδ T-cell auto-activation, its mechanism and further effects remain to be elucidated.

## 4. Conclusions

A histiocyte-rich, granulomatous variant of γδ T-cell lymphoma is extremely rare. Its potentially misleading resemblance to inflammatory granulomatous conditions could pose a diagnostic pitfall in this already challenging condition. This variant may resemble granulomatous mycosis fungoides and granulomatous slack skin syndrome, but it has a distinct clinical outcome. The possibility of an association between EBV and an increased number of histiocytes in PCGD-TCL is not clear and remains to be elucidated.

**Author Contributions:** T.P.: writing—original draft preparation, investigation. B.R.S.: project administration, methodology, conceptualization, resources. D.M.M.: methodology. O.S.: writing—review and editing, project administration, methodology, conceptualization, resources. All authors have read and agreed to the published version of the manuscript.

**Funding:** This research received no external funding.

**Institutional Review Board Statement:** An Institutional Review Board Statement was not needed because the current case series describes <3 cases.

**Informed Consent Statement:** Informed consent was waived because consent was already obtained at the time of biopsy.

**Conflicts of Interest:** The authors declare no conflict of interest.

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
