# Peer review of "A Previously Unrecognized Granulomatous Variant of Gamma-Delta T-Cell Lymphoma"

_dermatopathology, doi:10.3390/dermatopathology8020027_

Round 1

Reviewer 1 Report

The authors reported about two patients with cutaneous T-cell lymphoma with gamma-delta phenotype. They mentioned a previously unrecognized granulomatous variant.

Both biopsies contain a high number of histiocytes, and the infiltrate composition is indeed a diagnostic challenge and many dd should be considered.

Case 1: You described that PET-CT scan reveals bilateral pulmonary nodules, splenomegalie,… Is it true, that this was the first staging after the development of the skin lesions? If this is the case, by definition, you should diagnose a secondary cutaneous infiltrate of a systemic lymphoma. He also presented b symptoms when he came to the hospital. Why do you think that the skin manifestations precede the systemic involvement?

Did you exclude an immunosuppression?

Can you please mention the percentage of CD68/CD163 positive cells? Please show also a picture of betaF1.

Case 2: EBV detection is unusual in gamma/delta T-cell lymphoma. Please comment on this. What about other dd with EBV positivity?

Very important in granulomatous CD8 infiltrates is an exclusion of an immunosuppression. Especially in this case, which was also positive for EBV. Was an immunosuppression excluded? HIV?

Which complication were the reason for the death?

Can you please mention the percentage of CD68 positive cells? Please show also pictures of betaF1 and TCR delta.

In general, did you exclude a vasculitis? E.g. ANCA? ANA? ENA?

Discussion:

Please consider the above-mentioned aspects. Maybe the title should be also modified. Please discuss the dd (page 4, line 137ff) more in detail. Please consider also EVB-positive entities.

The part about details gamma/delta T-cells should be significantly shortened (page 5, line 158 – 195)

Conclusion should be also modified.

Author Response

Dear reviewer, we would like to thank you for the valuable input to this case series. Below you will find replies to each comment. The changes are highlighted in the revised manuscript. We hope that you will find this manuscript acceptable for the Dermatopathology.

Respectfully,

Tatsiana Pukhalskaya, MD

Question:

Case 1: You described that PET-CT scan reveals bilateral pulmonary nodules, splenomegaly,… Is it true, that this was the first staging after the development of the skin lesions? If this is the case, by definition, you should diagnose a secondary cutaneous infiltrate of a systemic lymphoma.

Answer:

The diagnosis of cutaneous gamma-delta T –cell lymphoma was based on the patient’s cutaneous symptoms as the initial and dominant clinical feature as well as smaller degree of systemic  disease that developed years after initial cutaneous findings.   At the time of initial cutaneous presentation 4-years prior to our evaluation, imaging studies failed to reveal an ongoing systemic process. Bone marrow was free of the disease at the time of staging as well.

We have added this information to the text (highlighted in yellow)

“At that time, imaging studies failed to reveal an ongoing systemic process.”

Question:

He also presented b symptoms when he came to the hospital. Why do you think that the skin manifestations precede the systemic involvement?

Answer:

We believe B-symptoms occur in most patients with primary cutaneous gamma – delta T-cell lymphoma and in case 1 might not serve as a reliable sign of systemic involvement. In particular, the fever and generalized weakness occurred concurrently with patient’s cutaneous symptoms and in our opinion are most likely related to his cutaneous disease.

Question:

Did you exclude an immunosuppression?

Answer:

There was no evidence of an immunosuppressive state. HIV test was performed and negative.

Question:

Can you please mention the percentage of CD68/CD163 positive cells?

Answer:

The percentage of CD68/163 cells were 70%. We have added this information to the text (highlighted in yellow):

                CD68/163 cells represented around 70% of the malignant infiltrate.

Question:

Please show also a picture of betaF1.

Answer:

We have included the picture of BetaF1.

Question:  

Case 2: EBV detection is unusual in gamma/delta T-cell lymphoma. Please comment on this. What about other dd with EBV positivity?

Answer:

We have added the following information to the discussion section of our manuscript (highlighted yellow and pink in the text):

“IHC analysis in our case 2 showed half of malignant cells to stain with EBV. This is an uncommon observation in PCGD-TCL. Such finding further complicated the diagnostic dilemma and brought the necessity to exclude NK/T-cell lymphoma nasal type (ENKTL) as well as secondary cutaneous involvement by other variants of γδ T-cell lymphoma. Both ENKTL and non-cutaneous forms of γδ T-cell lymphoma are known to express EBV [24]. ENKTL commonly originates in the upper respiratory tract, but involvement of other organs is frequent, particularly skin, where disease might be primary [25]. Cutaneous lesions of ENKTL present as erythematous or violaceous plaques and nodules. The histology might closely resemble subcutaneous panniculitis T-cell lymphoma, mycosis fungoides as well as PCGD-TCL [25]. Nevertheless, neoplastic cells of ENKTL are characterized by NK-cell phenotype and lack of T-cell markers i.e. TCR-alpha, -beta, -gamma and -delta as well as markers for CD3, CD4, CD5 and CD8. CD56 and cytotoxic proteins (TIA-1, granzyme B and perforin) are positive in majority of ENKTL cases, although CD56 is not unique to this form of lymphoma. In case 2 the atypical lymphocytes were exclusively T-cells with strong expression of CD3, CD8, CD56, granzyme B, TIA-1 and TCR γ/δ and absent expression of CD7, a common marker of ENKTL. In addition, laboratory and PET data during the initial evaluation confirmed the cutaneous confinement of the disease, ruling out the possibility of its secondary spread to the skin.”

“Furthermore, literature search revealed a report of suspected PCGD-TCL with EBV expression [23, 28]. Therefore, we believe that our finding adds to the body of literature.  With the increasing number of similar reports, it might be feasible in the future to separate the abovementioned malignancy as EBV T-cell lymphoproliferative disorder with gamma/delta phenotype In addition, the possibility of association between EBV and increased number of histiocytes in PCGD-TCL is not clear and remains to be elucidated”.

Question:

Very important in granulomatous CD8 infiltrates is an exclusion of an immunosuppression. Especially in this case, which was also positive for EBV. Was an immunosuppression excluded? HIV?

Answer:

Unfortunately, HIV-status of this patient was never determined. She suffered from type II diabetes, which can weaken the immune system. From patient’s available history, there was no documentation or evidence of an immunosuppressive state.

Question:

Which complication were the reason for the death?

Answer:

Autopsy was not performed for this patient and therefore we cannot comment on the precise cause of death as well as the status of patient’s lymphoma at the time of death. In summary, this patient’s lymphoma did not respond to any type of treatment. She developed 6 cm non-healing wound at the site of primary lesion complicated by secondary infection as well as numerous cutaneous lymphoma nodules all over her body.  Her lymph nodes showed signs of involvements as well.  The patient also suffered from severe thrombocytopenia. She ultimately requested palliative care, was admitted to hospice where she shortly passed away.

Question:

Can you please mention the percentage of CD68 positive cells?

Answer:

We have added the following information to the manuscript (highlighted yellow in the text):

“CD68 positive cells represented 70-80% of the malignant infiltrate”.

Question:

Please show also pictures of betaF1 and TCR delta.

Answer:

BetaF1 and TCR delta stains for case 2 were performed in the outside institution and unfortunately no longer available.

Question:

In general, did you exclude a vasculitis? E.g. ANCA? ANA? ENA?

Answer:

Patient in case 1 had originally received a diagnosis of a connective tissue disease- lupus panniculitis. As a result, he underwent connective tissue evaluation including ANA, ANCA and ENA. The ANA was positive while the ANCA and ENA were negative. In addition, there was no histologic evidence to support a diagnosis of any of the ANCA-positive vasculitides.  ANCA, ANA and ENA were not tested in Case #2.

Question:

Discussion:

Please consider the above-mentioned aspects. Maybe the title should be also modified.

Answer:

Thank you so much for the suggestion to modify the title. Could you kindly recommend the changes to be made in the title? We will appreciate your further input and directions.

Question:

Please discuss the dd (page 4, line 137ff) more in detail. Please consider also EVB-positive entities.

Answer:

We have modified the discussion to incorporate the differential diagnoses from the answers to the reviewer’s comments above (highlighted yellow in the text):

“IHC analysis in our case 2 showed half of malignant cells to stain with EBV. This is an uncommon observation in PCGD-TCL. Such finding further complicated the diagnostic dilemma and brought the necessity to exclude NK/T-cell lymphoma nasal type (ENKTL) as well as secondary cutaneous involvement by other variants of γδ T-cell lymphoma. Both ENKTL and non-cutaneous forms of γδ T-cell lymphoma are known to express EBV [24]. ENKTL commonly originates in the upper respiratory tract, but involvement of other organs is frequent, particularly skin, where disease might be primary [25]. Cutaneous lesions of ENKTL present as erythematous or violaceous plaques and nodules. The histology might closely resemble subcutaneous panniculitis T-cell lymphoma, mycosis fungoides as well as PCGD-TCL [25]. Nevertheless, neoplastic cells of ENKTL are characterized by NK-cell phenotype and lack of T-cell markers i.e. TCR-alpha, -beta, -gamma and -delta as well as markers for CD3, CD4, CD5 and CD8. CD56 and cytotoxic proteins (TIA-1, granzyme B and perforin) are positive in majority of ENKTL cases, although CD56 is not unique to this form of lymphoma. In case 2 the atypical lymphocytes were exclusively T-cells with strong expression of CD3, CD8, CD56, granzyme B, TIA-1 and TCR γ/δ and absent expression of CD7, a common marker of ENKTL. In addition, laboratory and PET data during the initial evaluation confirmed the cutaneous confinement of the disease, ruling out the possibility of its secondary spread to the skin.”

“Furthermore, literature search revealed a report of suspected PCGD-TCL with EBV expression [23, 28]. Therefore, we believe that our finding adds to the body of literature.  With the increasing number of similar reports, it might be feasible in the future to separate the abovementioned malignancy as EBV T-cell lymphoproliferative disorder with gamma/delta phenotype In addition, the possibility of association between EBV and increased number of histiocytes in PCGD-TCL is not clear and remains to be elucidated”.

Question:

The part about details gamma/delta T-cells should be significantly shortened (page 5, line 158 – 195)

Answer:

We have shortened the part discussing gamma/delta T-cells (highlighted yellow in the text):

                “The rarity of PCGD-TCL can be correlated with the cell of origin, which represents between 0.5% and 16% of the body’s T cells [28]. There are three Vδ segments, but the vast majority of γδ T cells express either the Vδ1 or Vδ2 T cell receptors, creating two distinguished subtypes of these cells: Vδ1 and Vδ2 [29]. Regarding types of γδ T cells found in the skin, recent findings by Jay Daniels et al. report PCGD-TCL to be derived from both Vδ1 and Vδ2 cells [30]. They proposed that γδ1 T-cells predominate in the epidermis and dermis and give rise to lymphomas originating in the epidermis and dermis, while panniculitis-based lymphomas arise from Vδ2 cells. In case 1, the tumor appeared as an epidermotropic, dermal and subcutaneous infiltrate, while in case 2, in was predominantly panniculitic in distribution. The difference in the initial clinical presentation (multiple subcutaneous nodules/plaques, constitutional symptoms and slow  course of the disease in case 1 compared to  single subcutaneous nodule without other symptoms and fulminant progression in case 2) might support possibility that these tumors originate from different δT-cell subtypes.”

Question:

Conclusion should be also modified.

Answer:

We have modified the conclusions to include the EBV statement (highlighted yellow in the text):

                “The possibility of association between EBV and increased number of histiocytes in PCGD-TCL is not clear and remains to be elucidated.”

Reviewer 2 Report

Abstract line13: this part of the sentence needs clarification "and following extensive workup and initial presumptive diagnoses of autoimmune processes were eventually diagnosed with PCGD-TCL . . ."

Abstract line 22 and Conclusion line 209:  see Caudron et al Dermatology. 2011;222(4):297-303.   Furthermore, on line 140 (Discussion), the authors mention a case of PCGTCL with granuloma formation.  Sentence should be removed or revised to state that this pattern is rare (previously reported).

Abstract line 25:  "This variant is analogous to granulomatous MF or GSS" This statement should be clarified because GSS is an indolent variant that does not clinically resemble gamma delta TCL.    This could be rephrased to state that the histologic features of granulomatous gamma delta CTCL may resemble granulomatous MF or GSS, but has a distinct clinical outcome.

Line 31:  The intent of the statement is evident, but there is some repetition (complex and complexity).  I recommend: . . . clinical and pathologic heterogeneity and diagnostic complexity".

Line 42:  This sentence should be modified as there are indolent variants of PCDGTCL (and gamma delta MF) that can be responsive to treatment.   See Khallaayoune et al Acta Derm Veneraol 2020 Jan 23;100(1) (one of 2 cases responded to tx).

Line 49: This sentence states: . . . "the malignant cells to express CD3+CD4-  . . ."  Please modify the sentence to remove the contradiction: "the malignant cells to typically have the following phenotype: " or remove the CD4- and the +'s from the CD markers.

Figure 3:  Please provide higher magnification images of immunostains to match the magnification in part 1.

Case 2:  The presence of extensive EBV expression in the tumor is concerning that this case is a distinct entity and it is difficult to exclude NK-cell lymphoma.  The presence of TCR gamma or delta expression by the tumor and the presence of T-cell clonality supports the proposed classification of a T-cell lymphoma as well the lack of angioinvasion typically seen in NK/T-cell lymphoma, but I recommend adding a sentence to state this supporting evidence and acknowledge the controversial nature so that is communicated clearly clearly to the audience.  Citation:  Y et al J Cut Pathol 2013 Mar; 40(3):310-6.

Conclusion needs to be reworded as it repeats statements from the Abstract that need modification, see above comments for Abstract line 22 and line 25. 

Remove hyphens within words in lines 210 and 211.

Author Response

Dear reviewer, we would like to thank you for the valuable input to this case series. Below you will find replies to each comment. The changes are highlighted in the revised manuscript. We hope that you will find this manuscript acceptable for the Dermatopathology.

Respectfully,

Tatsiana Pukhalskaya, MD

Question:

Abstract line13: this part of the sentence needs clarification "and following extensive workup and initial presumptive diagnoses of autoimmune processes were eventually diagnosed with PCGD-TCL . . ."

Answer:

We have clarified the abstract. Changes are highlighted green in the text:

                “We present a series of two patients who presented with firm subcutaneous nodules and were diagnosed with PCGD-TCL.”

Question:

Abstract line 22 and Conclusion line 209:  see Caudron et al Dermatology. 2011;222(4):297-303.  

Answer:

We appreciate the reviewer providing the reference we missed during our literature search. We have modified the abstract and conclusions. Changes are highlighted green in the text:

                “A histiocyte-rich, granulomatous variant of γδ T-cell lym-phoma is extremely rare.”

                “A histiocyte-rich, granulomatous variant of γδ T-cell lymphoma is extremely rare.”

Question:

Furthermore, on line 140 (Discussion), the authors mention a case of PCGTCL with granuloma formation.  Sentence should be removed or revised to state that this pattern is rare (previously reported).

Answer:

We have modified the discussion part to incorporate the report by Cudron et al. (highlighted green in the text):

                “Regarding PCGD-TCL, Caudron at al. described granulomatous infiltrate in a case of PCGT-TCL that was initially misdiagnosed as an inflammatory panniculitis [22].”

Question:

Abstract line 25:  "This variant is analogous to granulomatous MF or GSS" This statement should be clarified because GSS is an indolent variant that does not clinically resemble gamma delta TCL.    This could be rephrased to state that the histologic features of granulomatous gamma delta CTCL may resemble granulomatous MF or GSS, but has a distinct clinical outcome.

Answer:

We appreciate the reviewer’s suggestions. We have clarified the abstract by incorporating the statement above (highlighted green in the text)”

                “This variant may resemble granulomatous mycosis fungoides and granulomatous slack skin syndrome, but has a distinct aggressive clinical outcome”

Question:

Line 31:  The intent of the statement is evident, but there is some repetition (complex and complexity).  I recommend: . . . clinical and pathologic heterogeneity and diagnostic complexity".

Answer:

We appreciate this suggestion; we have incorporate the proposed statement in our manuscript (highlighted green in the text):

                “Primary cutaneous γδ T-cell lymphoma (PCGD-TCL) is an extremely rare and aggressive T-cell neoplasm with clinical and pathologic heterogeneity and diagnostic complexity [1].”

Question:

Line 42:  This sentence should be modified as there are indolent variants of PCDGTCL (and gamma delta MF) that can be responsive to treatment.   See Khallaayoune et al Acta Derm Veneraol 2020 Jan 23;100(1) (one of 2 cases responded to tx).

Answer:

We appreciate the provided data; we have incorporated this information into our manuscript (highlighted in green):

                “PCGD-TCL is commonly resistant to all forms of treatment, although cases with indolent clinical course have been reported [1, 10, 11].” 

Question:

Line 49: This sentence states: . . . "the malignant cells to express CD3+CD4-  . . ."  Please modify the sentence to remove the contradiction: "the malignant cells to typically have the following phenotype: " or remove the CD4- and the +'s from the CD markers.

Answer:

We have incorporated the reviewer’s suggestion into our manuscript (highlighted green in the text):

                “In most cases, immunohistochemical studies show the malignant cells to typically have the following phenotype:  CD3+, CD4-, CD56+, EBV -and variably  CD8+ [15].”

Question:

Figure 3:  Please provide higher magnification images of immunostains to match the magnification in part 1.

Answer:

Unfortunately, these stains are no longer available to us.

Question:

Case 2:  The presence of extensive EBV expression in the tumor is concerning that this case is a distinct entity and it is difficult to exclude NK-cell lymphoma.  The presence of TCR gamma or delta expression by the tumor and the presence of T-cell clonality supports the proposed classification of a T-cell lymphoma as well the lack of angioinvasion typically seen in NK/T-cell lymphoma, but I recommend adding a sentence to state this supporting evidence and acknowledge the controversial nature so that is communicated clearly clearly to the audience.  Citation:  Y et al J Cut Pathol 2013 Mar; 40(3):310-6.

Answer:

We appreciate the reviewer bringing up this point. We have incorporated the discussion regarding EBV in our manuscript (highlighted yellow and pink in the text):

“IHC analysis in our case 2 showed half of malignant cells to stain with EBV. This is an uncommon observation in PCGD-TCL. Such finding further complicated the diagnostic dilemma and brought the necessity to exclude NK/T-cell lymphoma nasal type (ENKTL) as well as secondary cutaneous involvement by other variants of γδ T-cell lymphoma. Both ENKTL and non-cutaneous forms of γδ T-cell lymphoma are known to express EBV [24]. ENKTL commonly originates in the upper respiratory tract, but involvement of other organs is frequent, particularly skin, where disease might be primary [25]. Cutaneous lesions of ENKTL present as erythematous or violaceous plaques and nodules. The histology might closely resemble subcutaneous panniculitis T-cell lymphoma, mycosis fungoides as well as PCGD-TCL [25]. Nevertheless, neoplastic cells of ENKTL are characterized by NK-cell phenotype and lack of T-cell markers i.e. TCR-alpha, -beta, -gamma and -delta as well as markers for CD3, CD4, CD5 and CD8. CD56 and cytotoxic proteins (TIA-1, granzyme B and perforin) are positive in majority of ENKTL cases, although CD56 is not unique to this form of lymphoma. In case 2 the atypical lymphocytes were exclusively T-cells with strong expression of CD3, CD8, CD56, granzyme B, TIA-1 and TCR γ/δ and absent expression of CD7, a common marker of ENKTL. In addition, laboratory and PET data during the initial evaluation confirmed the cutaneous confinement of the disease, ruling out the possibility of its secondary spread to the skin.”

“Furthermore, literature search revealed a report of suspected PCGD-TCL with EBV expression [23, 28]. Therefore, we believe that our finding adds to the body of literature.  With the increasing number of similar reports, it might be feasible in the future to separate the abovementioned malignancy as EBV T-cell lymphoproliferative disorder with gamma/delta phenotype In addition, the possibility of association between EBV and increased number of histiocytes in PCGD-TCL is not clear and remains to be elucidated”.

Question:

Conclusion needs to be reworded as it repeats statements from the Abstract that need modification, see above comments for Abstract line 22 and line 25.

Answer:

We have modified the conclusions accordingly (highlighted green in the text):

                “A histiocyte-rich, granulomatous variant of γδ T-cell lymphoma is extremely rare.”

“This variant may resemble granulomatous mycosis fungoides and granulomatous slack skin syndrome, but has a distinct aggressive clinical outcome.”

Question:

Remove hyphens within words in lines 210 and 211.

Answer:

We have removed hyphens in the referenced words (highlighted green in the text):

                “Due to its potentially misleading resemblance to inflammatory granulomatous conditions, it could pose a diagnostic pitfall in this already challenging condition.”

Reviewer 3 Report

The study of Pukhalskaya et al. is well-written and describes properly two cases of a variant of gamma-delta T-cell lymphoma. The quality of presentation is high and the interest to the readers is high. I do not have additional comments.

Author Response

Dear reviewer, we would like to thank you for your time to review this case series as well as your valuable feedback. We have made several changes in the manuscript to reflect the suggestions of other reviewers. We hope that you will find this manuscript acceptable for the Dermatopathology.

Respectfully,

Tatsiana Pukhalskaya, MD

Reviewer 4 Report

The authors report two interesting cases of gamma/delta T-cell lymphoma with prominent histiocytic and granulomatous component, respectively.

Comments:

  1. Page 2, line 52: The authors state that staining for TCR delta by immunohistochemistry can be performed on formalin-fixed paraffin-embedded tissue sections. The same is true for TCR gamma. Thus, both should be mentioned. It could be added that the staining for TCR delta gives more reliable results then the one for TCR gamma.
  2. Page 2, line 75: Adnexotropism should be specified whether it refers to syringotropism and/or folliculotropism.
  3. Page 2, line 85: As the patient shows systemic involvement at the time of diagnosis by definition despite the prolonged course the term cutaneous gamma/delta T-cell lymphoma is critical. It should be discussed that in this patient it cannot be distinguished with certainty whether the lesions represent cutaneous involvement in the context of a systemic gamma/delta T-cell lymphoma or a primary cutaneous gamma/delta T-cell lymphoma with secondary systemic spread during disease course.
  4. Page 3, case 2, line 104: It should be specified whether the cells express TCR gamma or TCR delta or both.
  5. Page 3, line 105: As there was detection of EBV in half of the tumor cells which is a very unusual finding for cutaneous gamma/delta T-cell lymphoma the diagnostic categorization of this case remains controversial. To my opinion it would better be classified as a EBV T-cell lymphoproliferative disorder with delta phenotype. This point needs to be discussed in detail in the discussion.
  6. Page 4, line 140: Typographical error: Scarabello instead of Scarabelo.
  7. Discussion, page 5: The authors in detail discuss the two subsets of gamma/delta T-cells and their immunology with a focus on cytokines. This is general information and does not particularly refer to the two cases. Instead the discussion should focus much more on the two cases and their particular findings instead of general information.

Author Response

Dear reviewer, we would like to thank you for the valuable input to this case series. Below you will find replies to each comment. The changes are highlighted in the revised manuscript. We hope that you will find this manuscript acceptable for the Dermatopathology.

Respectfully,

Tatsiana Pukhalskaya, MD

Question 1:

Page 2, line 52: The authors state that staining for TCR delta by immunohistochemistry can be performed on formalin-fixed paraffin-embedded tissue sections. The same is true for TCR gamma. Thus, both should be mentioned. It could be added that the staining for TCR delta gives more reliable results then the one for TCR gamma.

Answer:

We have respectfully incorporated this suggestion into our manuscript (highlighted pink in the text):

                “Staining for TCR-δ and TCR-γ by immunoperoxidase techniques can be conveniently performed in formalin-fixed paraffin-embedded tissue sections. [15]. The staining for TCR-δ gives more reliable results then the one for TCR-γ.”

Question 2:

Page 2, line 75: Adnexotropism should be specified whether it refers to syringotropism and/or folliculotropism.

Answer:

We have specified the syringotropism. We have added this information to the text (highlighted in pink):

                “There was notable angiodestruction, necrosis and syringotropism”

Question 3:

Page 2, line 85: As the patient shows systemic involvement at the time of diagnosis by definition despite the prolonged course the term cutaneous gamma/delta T-cell lymphoma is critical. It should be discussed that in this patient it cannot be distinguished with certainty whether the lesions represent cutaneous involvement in the context of a systemic gamma/delta T-cell lymphoma or a primary cutaneous gamma/delta T-cell lymphoma with secondary systemic spread during disease course.

Answer:

The diagnosis of cutaneous gamma-delta T –cell lymphoma was based on the patient’s cutaneous symptoms as the initial and dominant clinical feature as well as smaller degree of systemic  disease that developed years after initial cutaneous findings.   At the time of initial cutaneous presentation 4-years prior to our evaluation, imaging studies failed to reveal an ongoing systemic process. Bone marrow was free of the disease at the time of staging as well.

We have added this information to the text (highlighted in yellow)

“At that time, imaging studies failed to reveal an ongoing systemic process.”

Question 4:

Page 3, case 2, line 104: It should be specified whether the cells express TCR gamma or TCR delta or both.

Answer:

For case #1 TCR-delta was used, while in case #2 it was TCR-gamma. We have made appropriate corrections in the text (highlighted in pink).

Question 5:

Page 3, line 105: As there was detection of EBV in half of the tumor cells which is a very unusual finding for cutaneous gamma/delta T-cell lymphoma the diagnostic categorization of this case remains controversial. To my opinion it would better be classified as a EBV T-cell lymphoproliferative disorder with delta phenotype. This point needs to be discussed in detail in the discussion.

Answer:

Thank you so much for pointing this out. Several other reviewers have also advised to discuss this finding. We have incorporated all suggestions in the discussion section of our manuscript (highlighted yellow and pink in the text).

“IHC analysis in our case 2 showed half of malignant cells to stain with EBV. This is an uncommon observation in PCGD-TCL. Such finding further complicated the diagnostic dilemma and brought the necessity to exclude NK/T-cell lymphoma nasal type (ENKTL) as well as secondary cutaneous involvement by other variants of γδ T-cell lymphoma. Both ENKTL and non-cutaneous forms of γδ T-cell lymphoma are known to express EBV [24]. ENKTL commonly originates in the upper respiratory tract, but involvement of other organs is frequent, particularly skin, where disease might be primary [25]. Cutaneous lesions of ENKTL present as erythematous or violaceous plaques and nodules. The histology might closely resemble subcutaneous panniculitis T-cell lymphoma, mycosis fungoides as well as PCGD-TCL [25]. Nevertheless, neoplastic cells of ENKTL are characterized by NK-cell phenotype and lack of T-cell markers i.e. TCR-alpha, -beta, -gamma and -delta as well as markers for CD3, CD4, CD5 and CD8. CD56 and cytotoxic proteins (TIA-1, granzyme B and perforin) are positive in majority of ENKTL cases, although CD56 is not unique to this form of lymphoma. In case 2 the atypical lymphocytes were exclusively T-cells with strong expression of CD3, CD8, CD56, granzyme B, TIA-1 and TCR γ/δ and absent expression of CD7, a common marker of ENKTL. In addition, laboratory and PET data during the initial evaluation confirmed the cutaneous confinement of the disease, ruling out the possibility of its secondary spread to the skin.”

“Furthermore, literature search revealed a report of suspected PCGD-TCL with EBV expression [23, 28]. Therefore, we believe that our finding adds to the body of literature.  With the increasing number of similar reports, it might be feasible in the future to separate the abovementioned malignancy as EBV T-cell lymphoproliferative disorder with gamma/delta phenotype In addition, the possibility of association between EBV and increased number of histiocytes in PCGD-TCL is not clear and remains to be elucidated”.

Question 6:

Page 4, line 140: Typographical error: Scarabello instead of Scarabelo.

Answer:

We have fixed the typographical error (highlighted pink in the text):

                “Scarabello at al. described cutaneous granulomas in subcutaneous “panniculitis-like” variant of T- cell lymphoma, with cells derived from gamma-delta origin, CD4-, CD8+, and CD56+ [19].”

Question 7:

Discussion, page 5: The authors in detail discuss the two subsets of gamma/delta T-cells and their immunology with a focus on cytokines. This is general information and does not particularly refer to the two cases. Instead the discussion should focus much more on the two cases and their particular findings instead of general information.

Answer:

We have shortened the referenced section. To further clarify this for the reviewer, in that part of the discussion we are attempting to converse about possible mechanisms permitting high number of histiocytes in the presented entity.